# Homeostatic control of stearoyl desaturase expression via patched-like receptor PTR-23 ensures the survival of *C. elegans* during heat stress

**Siddharth R. Venkatesh**[1]☯, **Ritika Siddiqui**[1]☯, **Anjali Sandhu**[1]☯, **Malvika Ramani**[1], **Isabel R. Houston**[2], **Jennifer L. Watts**[2]*, **Varsha Singh**[1]¤*

1 Department of Developmental Biology and Genetics, Indian Institute of Science, Bangalore, India, 2 School of Molecular Biosciences and Center for Reproductive Biology, Washington State University, Pullman, Washington, United States of America

☯ These authors contributed equally to this work.
¤ Current address: School of Life Sciences, University of Dundee, United Kingdom
* jwatts@wsu.edu (JLW); varsha@iisc.ac.in (VS)

**Data Availability Statement:** Source data is available at dryad at https://doi.org/10.5061/dryad.15dv41p3j.

## Abstract

Organismal responses to temperature fluctuations include an evolutionarily conserved cytosolic chaperone machinery as well as adaptive alterations in lipid constituents of cellular membranes. Using *C. elegans* as a model system, we asked whether adaptable lipid homeostasis is required for survival during physiologically relevant heat stress. By systematic analyses of lipid composition in worms during and before heat stress, we found that unsaturated fatty acids are reduced in heat-stressed animals. This is accompanied by the transcriptional downregulation of fatty acid desaturase enzymes encoded by *fat-1*, *fat-3*, *fat-4*, *fat-5*, *fat-6*, and *fat-7* genes. Conversely, overexpression of the Δ9 desaturase FAT-7, responsible for the synthesis of PUFA precursor oleic acid, and supplementation of oleic acid causes accelerated death of worms during heat stress. Interestingly, heat stress causes permeability defects in the worm's cuticle. We show that *fat-7* expression is reduced in the permeability defective collagen (PDC) mutant, *dpy-10*, known to have enhanced heat stress resistance (HSR). Further, we show that the HSR of *dpy-10* animals is dependent on the upregulation of PTR-23, a patched-like receptor in the epidermis, and that PTR-23 downregulates the expression of *fat-7*. Consequently, abrogation of *ptr-23* in wild type animals affects its survival during heat stress. This study provides evidence for the negative regulation of fatty acid desaturase expression in the soma of *C. elegans* via the non-canonical role of a patched receptor signaling component. Taken together, this constitutes a skin-gut axis for the regulation of lipid desaturation to promote the survival of worms during heat stress.

**Funding:** This work was partly supported by the Indo French Center for Advanced Scientific Research (Grant no IFCP/6503-L) awarded to VS. Funding to JLW was provided by the National Institutes of Health (R01GM13883). SRV and RS were supported by SRF fellowship from the Council for Scientific and Industrial Research (CSIR), India. The funders had no role in the study design, data collection and analysis, decision to publish, or preparation of the manuscript.

**Competing interests:** The authors have declared that no competing interests exist.

## Author summary

Temperature fluctuation is a major environmental stressor for all living organisms. Here, we describe a mechanism that allows multicellular organisms to coordinate heat stress response between different tissues to ensure better survival. We find that a receptor in the skin of the worm controls stress response in adults. The receptor regulates lipid metabolism to enhance cellular health and organismal survival during chronic heat stress.

## Introduction

All living cells experience and respond to temperature fluctuations by upregulating chaperones and fine-tuning their metabolism. Activation of the HSP70 family of cytosolic chaperones is a conserved mechanism operating both in prokaryotes and eukaryotes [1–3]. Although dissociation of chaperone HSP90 from heat shock transcription factor HSF1 was long believed to be the trigger for chaperone synthesis, several recent reports suggest that fluidity changes in the lipid bilayer and microdomains may be the first sensor of heat stress in the cells [4–6]. Control of membrane fluidity, believed to be essential for signaling via membrane-associated receptors, has been observed in both prokaryotes and eukaryotes [7]. Decreased levels of unsaturated fatty acids in the lipid bilayer have been observed during temperature upshift, consistent with their role in cell survival during prolonged heat stress [8]. However, molecular sensors and regulators of membrane fluidity during heat stress are poorly understood.

At physiological temperature, saturated fatty acids are densely packed, while unsaturated fatty acids are disorganized and loosely packed in the lipid bilayer. The degree of unsaturation is highest at lower temperatures and lowest at higher temperatures (both in the permissive range) for various organisms. High-temperature tolerant bacterial species show decreased unsaturated to saturated fatty acid ratio [8–10]. A similar effect is seen in bacteria, plants, vertebrates, and invertebrates upon heat stress [11–17]. Consistent with this, stabilization of membrane lipids by drugs such as hydroxylamine derivative bimoclomol or small heat-shock protein HSP17 can protect cells against heat stress [18]. In mice, heat acclimatization achieved by modulation of membrane lipid composition protects them from heat shock and ischemic insults to the heart [19]. These studies suggest that the molecular packing of lipids is intricately controlled to maintain membrane fluidity and retain function under physiological and stress conditions.

*C. elegans*, a poikilotherm, adapts to temperature alteration by modifying its physiology and behavior. Worms respond to temperature changes by triggering a flight or fight response. Flight response or thermotaxis has been shown to be primarily dependent on the function of three pairs of amphid sensory neurons- AFD, AWC, and ASI [20]. These neurons also regulate the survival of worms under chronic heat stress. *C. elegans* exhibits both cell-autonomous and non-cell-autonomous regulation of heat stress response. In a previous study, we have shown that AWC and ASI neurons regulate the survival of worms at higher temperatures via STR-2, a G protein-coupled receptor, by regulating lipid metabolism non-cell autonomously [21]. Modulation of heat shock factor HSF-1 in the nervous system systematically coordinates lipid metabolism in non-neuronal tissues via cGMP/TGF-beta signaling [22]. These studies indicate that lipid homeostasis can be regulated non-cell autonomously. Cell-autonomous control of lipid metabolism is orchestrated via transcription regulators such as SBP-1, NHR-49, HLH-30, MDT-15, etc. [23–30]. Activation of acetyl-CoA dehydrogenase ACDH-11 at higher temperatures is required to suppress fatty acid desaturase activity to maintain membrane fluidity [31]. Mutations in fatty acid desaturases

*fat-6*, *fat-7*, and elongase *elo-2* increase heat stress tolerance, whereas diets supplemented with unsaturated fats reduce heat stress tolerance in worms [12]. These examples suggest that appropriate regulation of lipid metabolism and fatty acid desaturation might be central to heat stress response and adaptation in *C. elegans*.

Lipid metabolism can be regulated in animals by many conserved signaling pathways. Some evidence suggests that hedgehog (Hh) signaling, required for developmental and physiological programming, is an important regulator of lipid metabolism [32–37]. Subsequently, lipid metabolism also modulates Hh signaling [32, 38, 39]. Although *C. elegans* has 3 patched receptors (PTCs) and 24 patched-like receptors (PTRs), it lacks Smoothened (SMO), a component of the canonical hedgehog signaling pathway [40]. Loss of PTC-3, a cholesterol transporter, reduces fatty acid desaturation [41], while the inhibition of PTR-24 leads to lipid accumulation in worms [42]. However, the roles of the majority of the patched receptors in heat stress response remain to be studied.

In this study, we examined the contribution of lipid desaturation in the heat stress response of *C. elegans*. We specifically asked whether unsaturated fatty acids such as oleic acid are deleterious for survival during chronic stress. Using a combination of fatty acid analyses and genetic approaches, we show that heat stress and cuticle permeability defect (PD) have a direct impact on the expression of the Δ9 fatty acid (stearoyl) desaturase FAT-7. This desaturase is necessary for the synthesis of the mono-unsaturated fatty acid (MUFAs), oleic acid. We find that desaturase expression is dependent on patched-like receptor PTR-23 in the epidermis of *C. elegans*. The receptor itself is upregulated during heat stress and in PD collagen mutants and suppresses FAT-7 to provide protection. Our study uncovers a PTR-23-fatty acid desaturase axis for the survival of worms during heat stress.

## Results

### Lipid composition in *C. elegans* is skewed in favor of saturated fatty acids during heat stress

Temperature fluctuation is a frequent environmental stressor for animals. To understand if chronic heat stress causes alteration in lipid composition in *C. elegans*, we studied the alterations in the levels of transcripts for various fatty acid desaturase enzymes in wild-type N2 worms (WT). For PUFAs, we examined *fat-1, -2, -3, -4*, and for MUFAs, *fat-5, -6, -7* desaturases (Fig 1A) [43–47], using qRT-PCR in control and heat-stressed worms (Fig 1B). We found that all desaturases, except *fat-2*, were significantly downregulated in heat-stressed worms (Fig 1C).

We confirmed the downregulation of palmitoyl desaturase *fat-5* and stearoyl desaturase *fat-7* using translational GFP reporters FAT-5::GFP (S1 Fig) and FAT-7::GFP (Figs 1D and 1E), respectively. Downregulation was observed in heat-stressed worms at both 4h and 8h for both FAT-7 and FAT-5 reporters (Figs 1D and 1E and S1 Fig). We also performed a temporal analysis on the expression of various enzymes involved in fatty acid metabolism in heat-stressed WT worms (S2A–S2I Fig). Notably, the transcript for diacylglycerol acyl transferase (*dgat-2*), the rate-limiting enzyme for neutral lipid synthesis, was downregulated in heat-stressed worms (S2H Fig). Similarly, the transcript for acyl CoA synthetase encoded by *acs-2*, necessary for fatty acid breakdown, was also downregulated in heat-stressed worms (S2I Fig). Our results suggest that while neutral lipid synthesis and fatty acid oxidation are suppressed during heat stress, desaturases are controlled intricately (S2A–S2I Figs). We also analyzed the expression of the cytosolic chaperone *hsp-16.2* and found that its transcripts were upregulated during heat shock, as expected (S2J Fig).

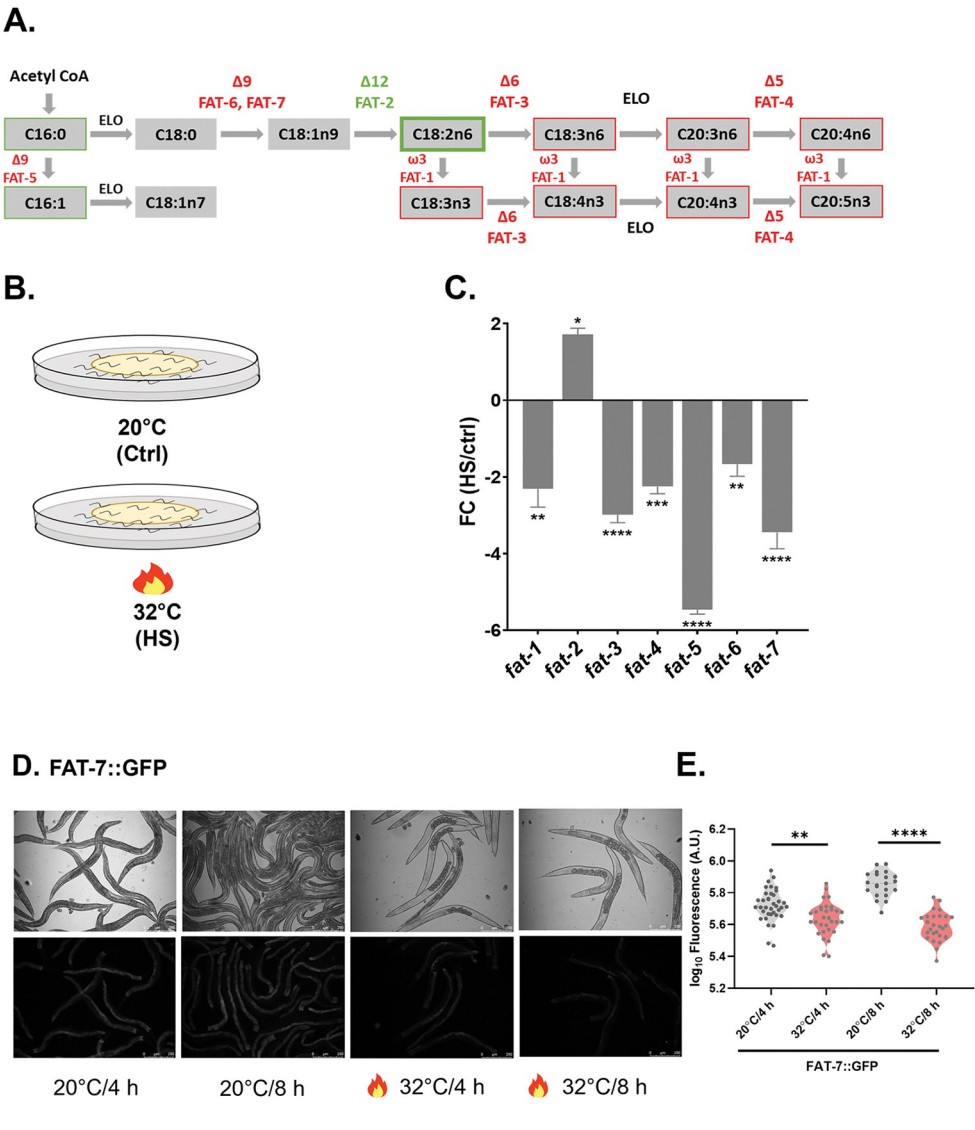

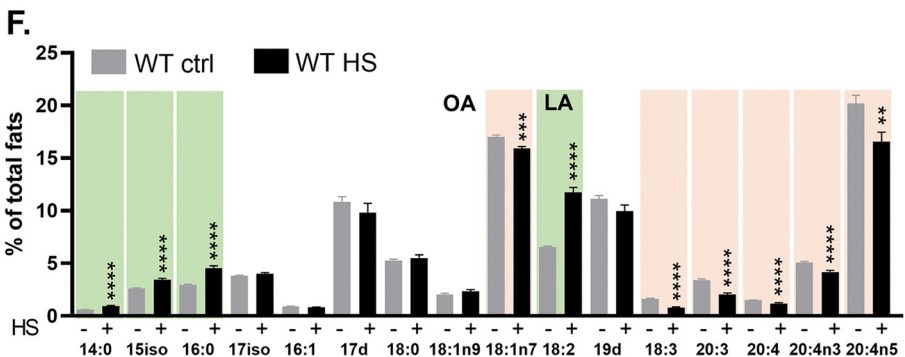

**Fig 1. Heat stress induces rapid adaptation in lipid composition in *C. elegans*.** (A) A schematic representation of fatty acid synthesis and fatty acid desaturation in *C. elegans*. Fatty acid desaturases significantly upregulated in heat-stressed (HS, 32˚C) worms are labeled green, and those significantly downregulated in HS worms are labeled red. Fatty acids significantly increased in HS worms over ctrl are presented within green boxes and those significantly reduced in HS worms are presented within peach boxes. (B) A schematic representation of our two test conditions. (C) qPCR

analysis of fatty acid desaturase genes in HS (32˚C) over ctrl (20˚C) WT animals. Statistical significance was calculated using unpaired Student's '$t$' test with Welch's correction. (C) qPCR analyses of the fatty acid desaturase genes in HS worms over ctrl worms. Statistical significance was calculated using Student's '$t$' test with Welch's correction. (D) FAT-7::GFP expression in ctrl and HS animals at indicated times and temperatures (scale bar, 250 μm) and their (E) quantification. Statistical significance for (E) was calculated using a post-hoc Dunnett test. ns (not significant), $P > 0.05$; * $P \leq 0.05$; ** $P \leq 0.01$; *** $P \leq 0.001$; **** $P \leq 0.0001$. (F) GC-MS analysis of fatty acids in wild-type N2 animals (WT) at 32˚C (heat-stressed, HS) or maintained at 20˚C (control, ctrl). Fatty acids significantly increased in HS worms over ctrl are shaded in green and those significantly reduced in HS worms are shaded in peach. Error bars represent SEM. Statistical significance was calculated using $t$-test. Also refer to S1 Table.

To validate if the changes in the desaturase genes translate to changes in lipid composition, we performed fatty acid analyses in WT worms maintained at 20˚C and under heat stress (shifted to 32˚C for 8 hours) by gas chromatography-mass spectrometry (GC-MS). We found that there were dramatic changes in the relative amounts of several fatty acids following heat stress, while a few remained unaltered (Fig 1F and S1 Table). There was a marked and significant increase (shaded in green in Fig 1A) in the levels of saturated fatty acids, myristic acid (C14:0), and palmitic acid (C16:0), while there was a decline (shaded in peach) in 6 out of 7 polyunsaturated fatty acids (PUFAs) (Fig 1F). Amongst mono-unsaturated fatty acids (MUFAs), there was no change in the levels of palmitoleic acid (C16:1) or oleic acid (C18:1n-9), and some decline in vaccenic acid (C18:1n-7). Overall, our experiments indicated that lowering of PUFAs by the downregulation of desaturase enzymes is an important component of worms' response to heat stress, most likely to retain membrane function and to ensure survival at higher temperatures.

## Excess oleic acid reduces the survival of worms during heat stress

A decline in the levels of PUFAs was observed in heat-stressed WT worms. To test whether the accumulation of PUFAs is deleterious during heat stress, we subjected worms supplemented with linoleic acid (LA) to survival at 32˚C. We observed that LA-supplemented worms did not show disparate survival from unsupplemented worms (Fig 2A). The decline in PUFAs was accompanied by a reduction in transcripts for the Δ9 desaturase FAT-7. This stearoyl desaturase is required for the synthesis of oleic acid (OA), the precursor for all PUFAs. We predicted that an increased level of oleic acid will be deleterious for the survival of worms during chronic heat stress. To test this, we studied the survival of WT worms supplemented with OA. As shown in Fig 2B, supplementation with OA enhanced the susceptibility of worms to heat stress. Additionally, we also checked the survival of a FAT-7 overexpression (OE) strain, DMS303, and the Δ12 desaturase *fat-2* mutants that show an accumulation in OA [47] during heat stress. As shown in Fig 2C–2D, FAT-7 OE and *fat*-2 mutation enhanced the susceptibility of worms to heat stress, suggesting that too much oleic acid was indeed deleterious for survival during chronic heat stress. As a control for heat stress, we also analyzed the survival of worms during hyperosmotic stress and found no difference in the survival phenotype of WT, FAT-7 OE, and OA-supplemented WT worms (S3A and S3C Fig). We also subjected WT, FAT-7 OE, and OA-supplemented WT strains to oxidative stress using 10 mM $H_2O_2$ and found no difference in their survival (S3B and S3D Fig). We also studied the effect of increased OA levels on desaturase expression. In both OA supplementation and FAT-7 OE animals, *fat-2* desaturase expression was enhanced, while *fat-7* expression was downregulated in OA supplemented worms (Fig 2E and 2F). These indicated that oleic acid does not have a broad impact on stress responses; rather, it selectively affects survival during heat stress, where optimization of membrane fluidity is known to be crucial.

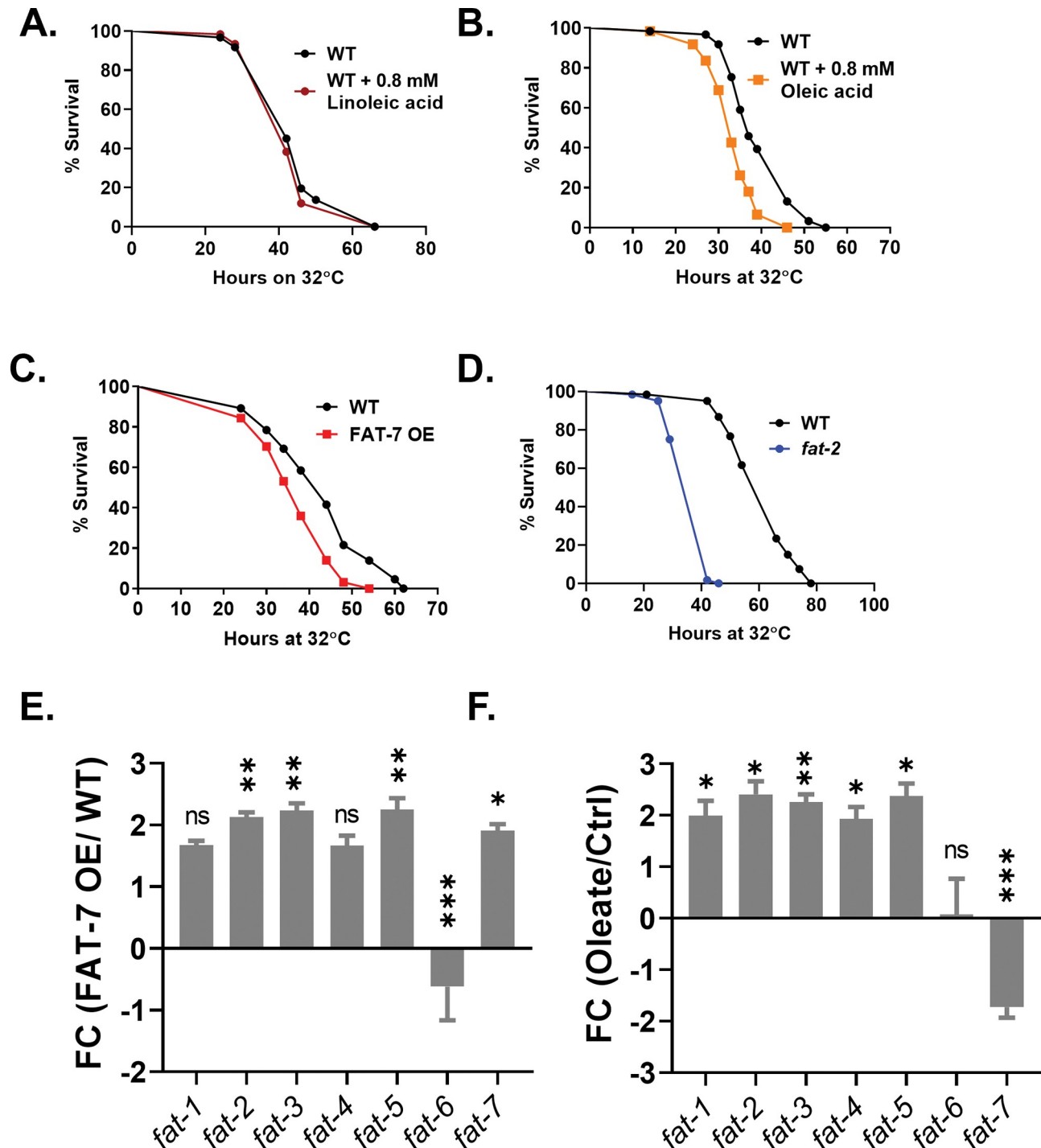

**Fig 2. Excess oleic acid is deleterious for survival during chronic heat stress.** (A) Kaplan-Meier survival curves of WT and 0.8 mM linoleic acid supplemented WT during heat stress at 32˚C (P = 0.5921; N = 25–30; n = 3). (B) Kaplan-Meier survival curves of WT and 0.8 mM oleic acid supplemented WT during heat stress at 32˚C (P<0.0001; N = 25–30; n = 3). (C) Kaplan-Meier survival curves of N2 and FAT-7 OE animals during heat stress at 32˚C (P = 0.0002; N = 25–30; n = 3). (D) Kaplan-Meier survival curves of N2 and *fat-2* animals during heat stress at 32˚C (P<0.0001; N = 25–30; n = 3). (E) qPCR analysis of fatty acid desaturase genes in FAT-7 OE animals at 20˚C. (F) qPCR analysis of fatty acid desaturase genes in oleic acid supplemented WT animals at 20˚C. ns (not significant), P > 0.05, * P ≤ 0.05, ** P ≤ 0.01, *** P ≤ 0.001, **** P ≤ 0.0001 as determined by unpaired Students '*t*' test with Welch's correction. Error bars represent SEM.

## Permeability-determining collagens regulate heat stress response and FAT-7 desaturase expression

We have previously reported that the permeability-determining (PD) collagen, DPY-10, necessary for maintaining the permeability barrier function of *C. elegans* cuticle, is linked to heat stress response (HSR) [48]. Notably, we observed nuclear staining by Hoechst stain in the head of 24 h heat-stressed worms but not the control worms (S4A–S4C Fig). This suggested cuticle permeability defects in heat-stressed worms. Consistent with our previous report, we found that mutations in *dpy-10* led to enhanced survival of animals during heat stress (Fig 3A). Based on the roles implicated for FAT-7 in heat stress resistance (Fig 2), we hypothesized that the heat stress resistance of PD collagen mutants is due to the reduced expression of FAT-7. To test this hypothesis, we analyzed the expression of transcripts of various desaturases in *dpy-10 (ves2003)* animals. Although there was a slight increase in the transcript levels for *fat-1* and *fat-4*, there was a dramatic downregulation of *fat-7* transcript (12-fold or more) (Fig 3B). FAT-7::GFP expression was also reduced in *dpy-10* mutants (Fig 3C and 3D), suggesting lower levels of OA.

To confirm the effect of *dpy-10* mutation on the synthesis of fatty acids, we performed GC-MS analyses of fatty acids in these animals. As shown in Fig 3E, two PUFAs and the MUFA OA (18:1n9) were reduced (shaded in peach), while saturated fatty acids palmitoleic acid and stearic acid (16:0 and 18:0) were increased in *dpy-10(ves2003)* compared to WT animals. An increase in saturated and a decrease in unsaturated fatty acids in *dpy-10* mutant is consistent with the enhanced HSR compared to wild-type animals.

## Patched-like receptor PTR-23 mediates heat stress resistance of PD collagen mutants

Animals carrying a mutation in the PD collagen *dpy-10* were not only heat stress resistant but also had significantly lower levels of transcript for FAT-7 desaturase, expressed predominantly in the intestine of worms. We sought to determine the link between cuticle collagen defect and lipid homeostasis in the soma. In a previous study on gene expression analyses in *dpy-9* and *dpy-10* mutant animals [49], several patched-like receptor (*ptr*) genes were found to be altered. To test the involvement of one or more of these receptors, we performed RNAi for 22 *ptr* encoding genes individually and scored for survival during heat stress at 32°C for 24 hours in *dpy-10*(ves2003) animals (Fig 4A). RNAi inhibition of *ptr-23* caused significant suppression of the enhanced HSR phenotype of the *dpy-10(ves2003)* mutant.

PTR-23 is a patched-like receptor expressed in the epidermis (skin) of *C. elegans* [50], the tissue expressing most collagens, including DPY-10 [51]. We found that the levels of *ptr-23* transcripts were upregulated in *dpy-10(ves2003)* over WT animals (Fig 4B). Importantly, *ptr-23(ok3663)* mutation disrupted the HSR phenotype of *dpy-10* mutant and RNAi animals (Figs 4C and S5A). Additionally, we also observed an HS-dependent upregulation in the *ptr-23* transcript levels in the WT animals (Fig 4D). As shown in Fig 4E, *ptr-23(ok3663)* animals were more susceptible to heat stress than WT animals but had wild-type susceptibility to osmotic stress and oxidative stress (S5B and S5C Fig). Since PTR-23 is known to be expressed in the epidermis, we examined the effect of targeted *ptr-23* RNAi in the epidermis (in NR222 strain) or in the intestine (VP303 strain) on heat stress survival and found that epidermal expression of PTR-23 was needed for survival upon heat stress and *fat-7* downregulation in WT animals (Figs 4F and 4G and S5D). Further, PTR-23 also had some influence on other desaturases (S5E Fig). Taken together, our experiments showed that HSR in wild-type and PD collagen mutants is dependent on patched-like receptor PTR-23 expressed in the epidermis of *C. elegans*. Given that heat stress affects cuticular integrity (S4A–S4C Fig), our results suggest a role for the

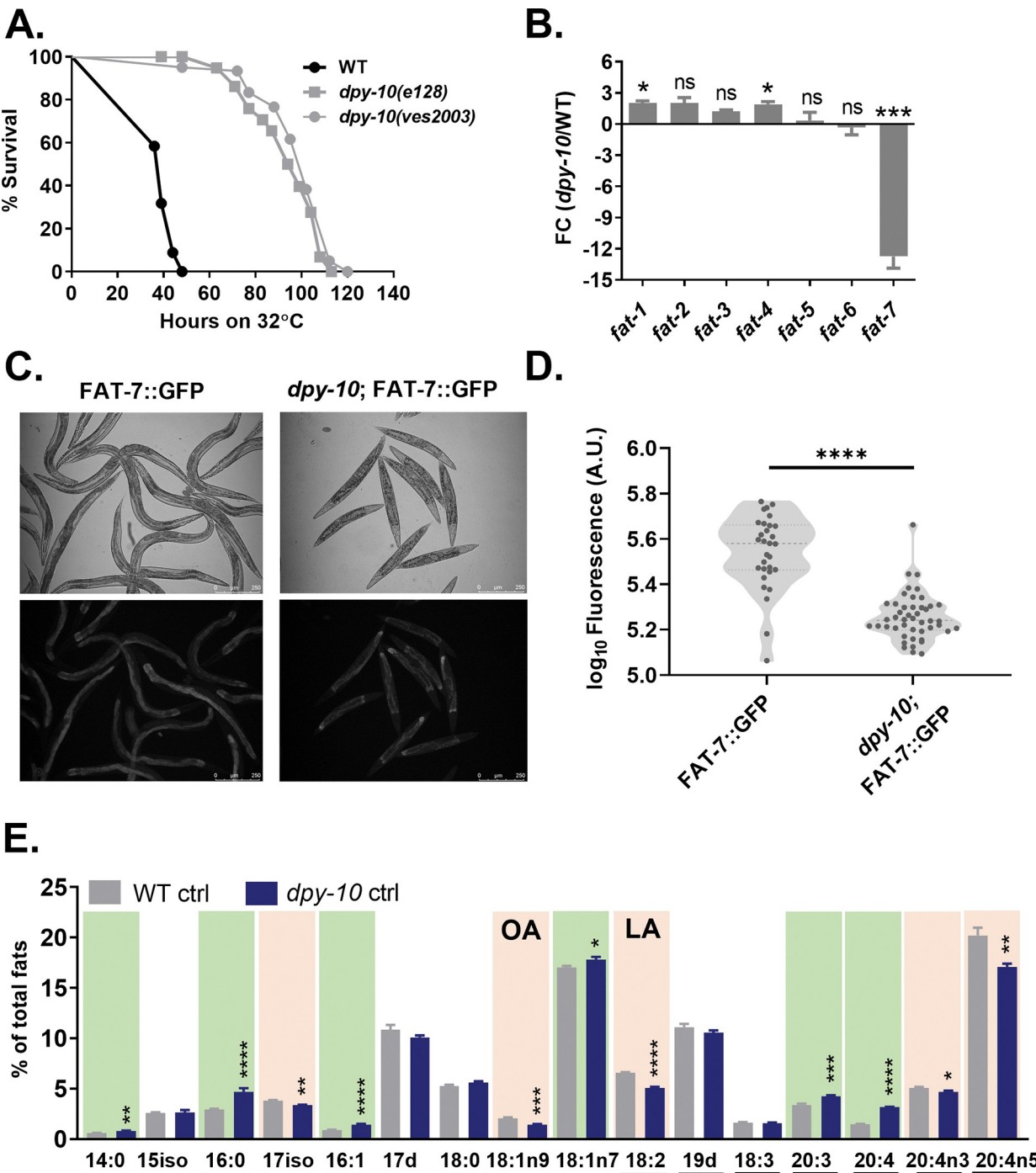

**Fig 3. Permeability-determining collagens regulate HSR and desaturase levels.** (A) Kaplan-Meier survival curves of WT, *dpy-10(e128)* and *dpy-10(ves2003)* animals during heat stress at 32˚C (P<0.0001; N = 25–30; n = 3). (B) qPCR analysis of fatty acid desaturase genes in *dpy-10 (ves2003)* over WT animals at 20˚C. (C) FAT-7::GFP expression in *dpy-10(ves2003)* and control animals at 20˚C and its (D) quantification. (E) GC-MS analysis of fatty acids in *dpy-10(ves2003)* and WT animals maintained at 20˚C. Fatty acids significantly increased in *dpy-10* animals are shaded in green and fatty acids significantly reduced are shaded in peach. The WT ctrl data in 3E is the same as WT ctrl data in Fig 1F. ns (not significant) P > 0.05, * P ≤ 0.05, ** P ≤ 0.01, *** P ≤ 0.001, **** P ≤ 0.0001 as determined by Students '*t*' test with Welch's correction. Error bars represent SEM.

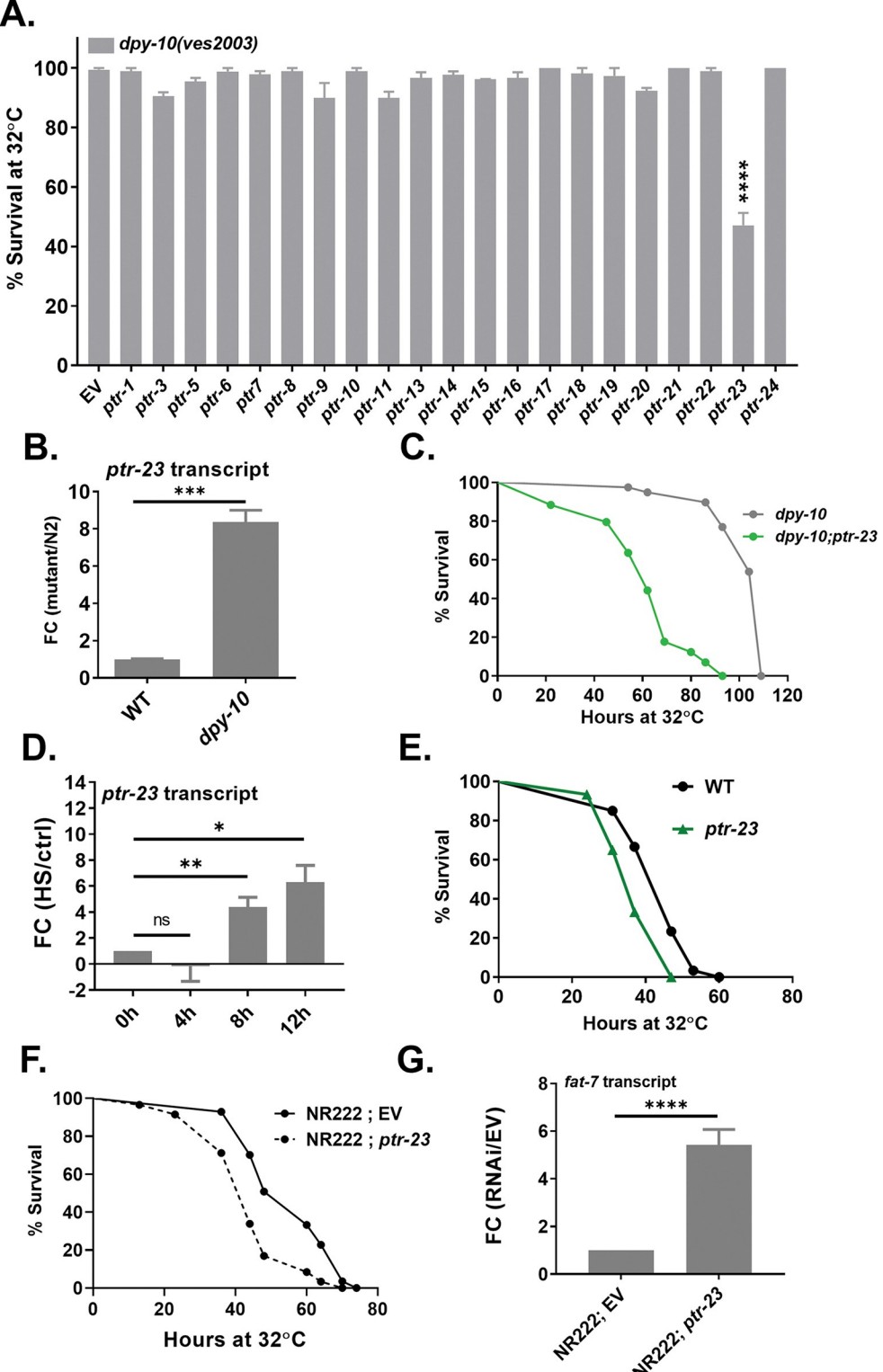

**Fig 4. PTR-23 regulates the heat stress resistance of *dpy-10* mutants.** (A) Percent survival of *dpy-10(ves2003)* animals with empty vector (EV) or *ptr* RNAi during heat stress at 32˚C for 24 hours. (B) Basal levels of *ptr-23* transcript in WT and *dpy-10(ves2003)* animals at 20˚C. (C) Kaplan-Meier survival curves of *dpy-10(ves2003)* and *dpy-10(ves2003); ptr-23(ok3663)* animals during heat stress at 32˚C (P<0.0001; N = 25–30; n = 3). (D) *ptr-23* transcript levels of WT worms heat-stressed at 32˚C for 4, 8, and 12 h. (E) Kaplan-Meier survival curves of WT *and ptr-23*

(ok3663) animals during heat stress at 32˚C (P<0.0001; N = 25–30; n = 3). (F) Kaplan-Meier survival curves of the epidermal RNAi NR222 (kzIs9 [(pKK1260) lin-26p::NLS::GFP + (pKK1253) lin-26p::rde-1 + rol-6(su1006)]) animals with EV and *ptr-23* RNAi during heat stress at 32˚C (P<0.0001; N = 25–30; n = 3). (G) Basal levels of *fat-7* transcript in NR222 animals with EV and *ptr-23* RNAi. ns (not significant), P > 0.05, * P ≤ 0.05, ** P ≤ 0.01, *** P ≤ 0.001, **** P ≤ 0.0001 as determined by unpaired Students '*t*' test with Welch's correction. Error bars represent SEM.

cuticle in regulating FAT-7 expression and, consequently, heat stress survival via the patched receptor PTR-23.

## PTR-23 is required for the repression of FAT-7 expression in PD collagen mutant

After identifying a PTR-23-mediated downregulation of *fat-7* and modulation of heat stress response in WT animals, we wanted to test if the HSR of *dpy-10* animals was due to the downregulation of the MUFA OA via PTR-23. We subjected OA-supplemented *dpy-10* worms to heat stress. OA-supplemented *dpy-10* animals were significantly susceptible to heat stress (Fig 5A).

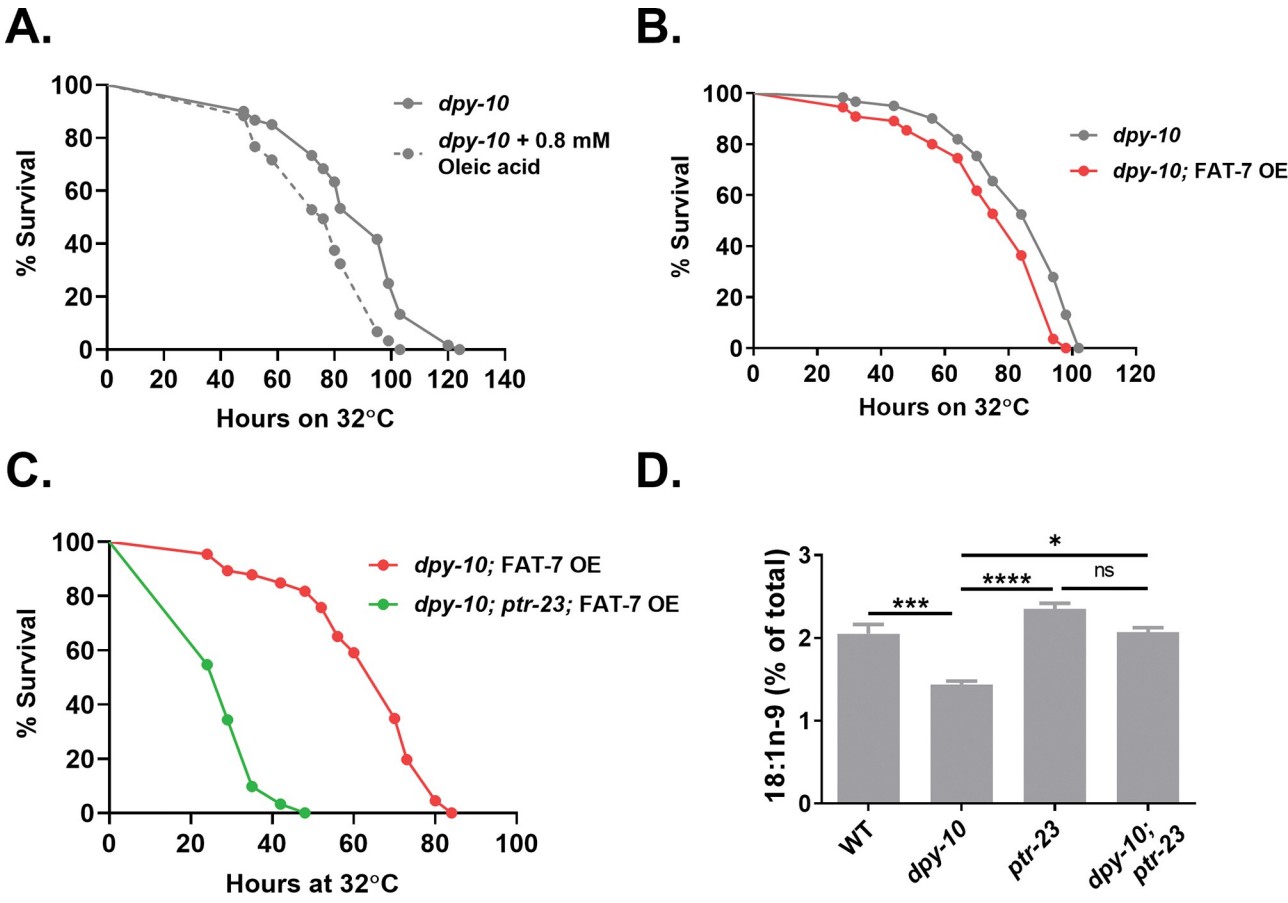

**Fig 5. Oleic acid supplementation or FAT-7 over-expression diminishes heat stress resistance of *dpy-10* animals.** (A) Kaplan-Meier survival curves of *dpy-10*(ves2003) and *dpy-10(ves2003)* animals supplemented with 0.8 mM oleic acid during heat stress at 32˚C (P<0.0001; N = 25–30; n = 3). (B) Kaplan-Meier survival curves of *dpy-10*(ves2003) and *dpy-10(ves2003); FAT-7 OE* animals during heat stress at 32˚C (P = 0.0017; N = 25–30; n = 3). (C) Kaplan-Meier survival curves of *dpy-10(ves2003); FAT-7 OE* and *dpy-10*(ves2003); *ptr-23; FAT-7 OE* animals during heat stress at 32˚C (P<0.0001; N = 25–30; n = 3). (D) Levels of oleic acid (C18:1n9), in WT, *dpy-10*(ves2003), *ptr-23* (ok3663) and *dpy-10; ptr-23*(ok3663) animals at 20˚C. ns (not significant), P > 0.05, * P ≤ 0.05, ** P ≤ 0.01, *** P ≤ 0.001, **** P ≤ 0.0001 as by post-hoc Tukey's test. Error bars represent SEM.

Consistent with this, *dpy-10* animals with FAT-7 OE were also susceptible to heat stress (Fig 5B). Furthermore, mutation in *ptr-23* exacerbated the susceptibility to heat stress in *dpy-10*; FAT-7 OE animals (Fig 5C). This suggested that PTR-23 might suppress the expression or the activity of FAT-7 desaturase. We also analyzed the levels of various fatty acids in *dpy-10 (ves2003)* and *dpy-10(ves2003); ptr-23(ok3663)* animals by GC-MS (S1 Table). Saturated fatty acid palmitic acid (C16:0) was higher in *dpy-10* than in WT animals, whereas levels of oleic acid and linoleic acid were reduced in *dpy-10* animals (S1 Table), consistent with its HSR phenotype. The reduction of oleic acid seen in *dpy-10* mutant (compared to WT) was suppressed in *dpy-10 (ves2003); ptr-23(ok3663)* animals, at the basal level as well as during heat stress (Fig 5D).

To study how PTR-23 impacts oleic acid levels, we studied the levels of fat-7 using FAT-7::GFP reporter in WT, *dpy-10(ves2003)*, *ptr-23(ok3663)* and *dpy-10(ves2003); ptr-23(ok3663)* animals (Fig 6A–6F) in control and heat-stressed conditions. As expected, FAT-7::GFP expression was lowered in WT and *dpy-10* animals upon heat stress (Fig 6A and 6B). However, heat stress did not downregulate FAT-7::GFP expression in *ptr-23* or *dpy-10; ptr-23* animals (Fig 6C and 6D), suggesting that PTR-23 activity was needed for the downregulation of *fat-7*. We also analyzed the expression of *fat-7* transcript in WT, *dpy-10(ves2003)*, and *dpy-10 (ves2003); ptr-23(ok3663)* animals at 20˚C. As shown in Fig 6G, the expression level of *fat-7* transcript was 30-fold lower in *dpy-10* animals. However, *fat-7* level was rescued in *dpy-10; ptr-23* compared to *dpy-10* animals. This suggested that PTR-23 is necessary to suppress the expression of *fat-7* transcript. Finally, we tested the expression of *fat-7* transcript in heat-stressed WT, *dpy-10(ves2003)*, and *dpy-10(ves2003); ptr-23(ok3663)* animals and found a similar regulation by *ptr-23* (Fig 6H). An increase in the transcription of *ptr-23* (Fig 4D) correlated with a decline in transcripts of *fat-7* (S2G Fig), providing additional evidence that PTR-23 is a negative regulator of *fat-7* expression in *C. elegans*.

Taken together, our study shows that lowering the relative levels of unsaturated fatty acids is an important component of the heat stress response in *C. elegans*. Mechanistically, it is accomplished by upregulation of the PTR-23-dependent pathway, which serves to downregulate the expression of fatty acid desaturase FAT-7. The patched receptor-desaturase axis operative in *C. elegans* soma (skin to gut) is essential for improving worm survival during heat stress (Fig 7).

## Discussion

Temperature fluctuation is a physiologically relevant stress experienced by all organisms, seasonally and diurnally. In this study, we found that an important component of the response to heat stress is the lowering of unsaturated fatty acid levels in *C. elegans*. We observed a coordinated decline in the transcription of genes encoding fatty acid desaturases such as FAT-7. During prolonged heat stress, upregulation of *fat-7* or exogenous supplementation of oleic acid (OA) causes a reduction in the survival of worms. We find that the expression of *fat-7* is controlled non-cell autonomously by a patched-like receptor, PTR-23, in the epidermis of *C. elegans*. PTR-23 expression is increased in response to heat stress and by disruption of the permeability barrier of the cuticle/skin in *C. elegans*. Overall, we show the presence of an epidermis-intestine axis to regulate fatty acid desaturase expression in response to heat stress.

In our study, we show that supplementation of the MUFA OA and the overexpression of FAT-7, the Δ9 desaturase that converts stearic acid (C18:0) to OA (C18:1n9), hampers the worms' survival during heat stress. Additionally, the *fat-2* mutants also showed a reduced survival, similar to the OA-supplemented and FAT-7 OE animals. FAT-2 is a Δ12 desaturase involved in the conversion of oleic acid to linoleic acid through desaturation [47]. These mutants have significantly lowered levels of PUFAs in their total lipid profiles compared to the

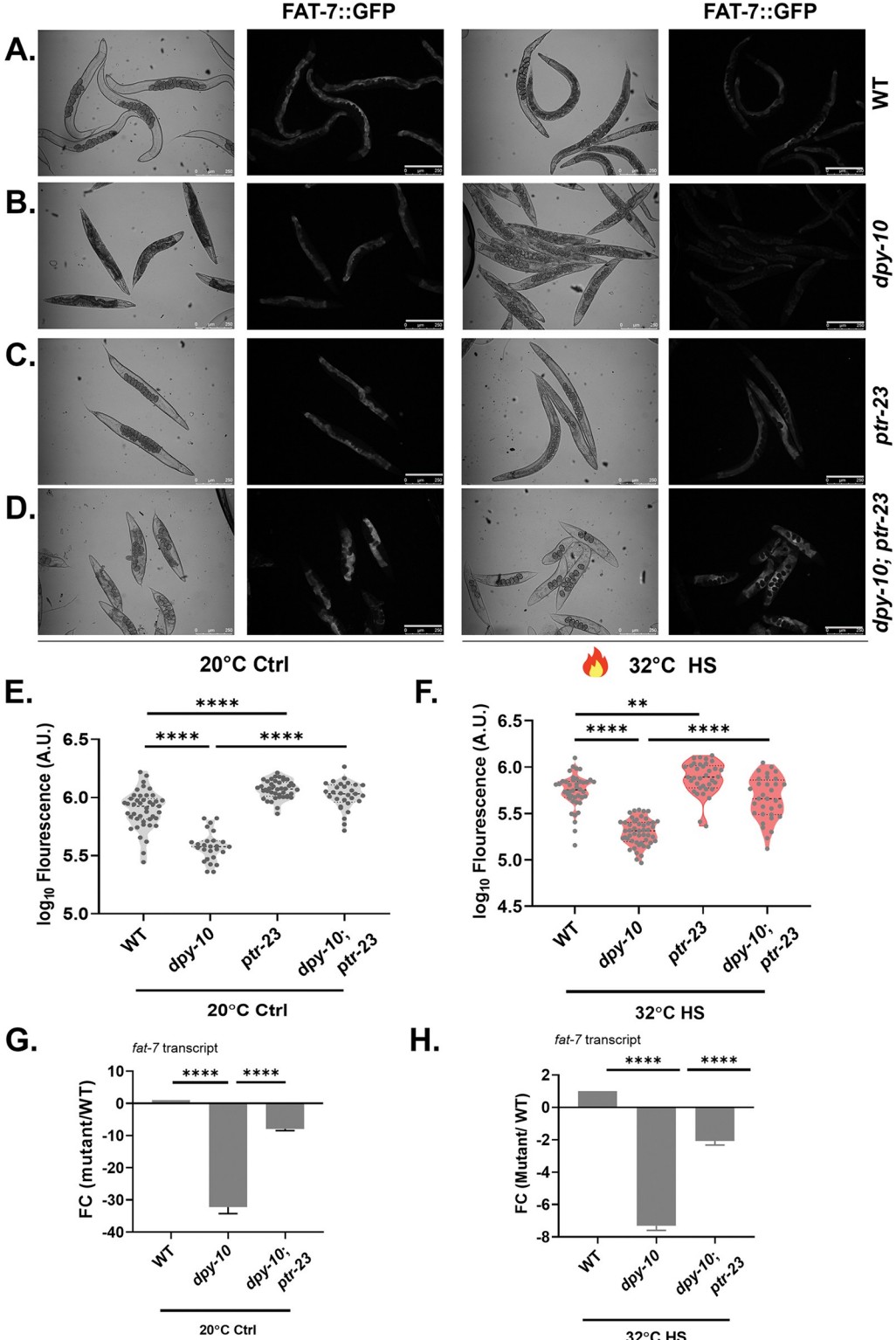

**Fig 6. PTR-23 regulates stearoyl desaturase FAT-7 levels in *dpy-10* animals.** FAT-7::GFP expression in (A) control, (B) *dpy-10(ves2003)*, (C) *ptr-23(ok3663)* and (D) *dpy-10(ves2003);ptr-23(ok3663)* animals at 20˚C or heat stressed at 32˚C for 8 hours (scale bar 250 μM). Quantification of GFP intensity at (E) 20˚C and (F) 32˚C. Levels of *fat-7* transcripts in WT, *dpy-10(ves2003)*, and *dpy-10(ves2003); ptr-23(ok3663)* animals at (G) 20˚C and 32˚C. ns (not significant), P > 0.05, * P ≤ 0.05, ** P ≤ 0.01, *** P ≤ 0.001, **** P ≤ 0.0001 as determined by unpaired Students '*t*' test. Error bars represent SEM.

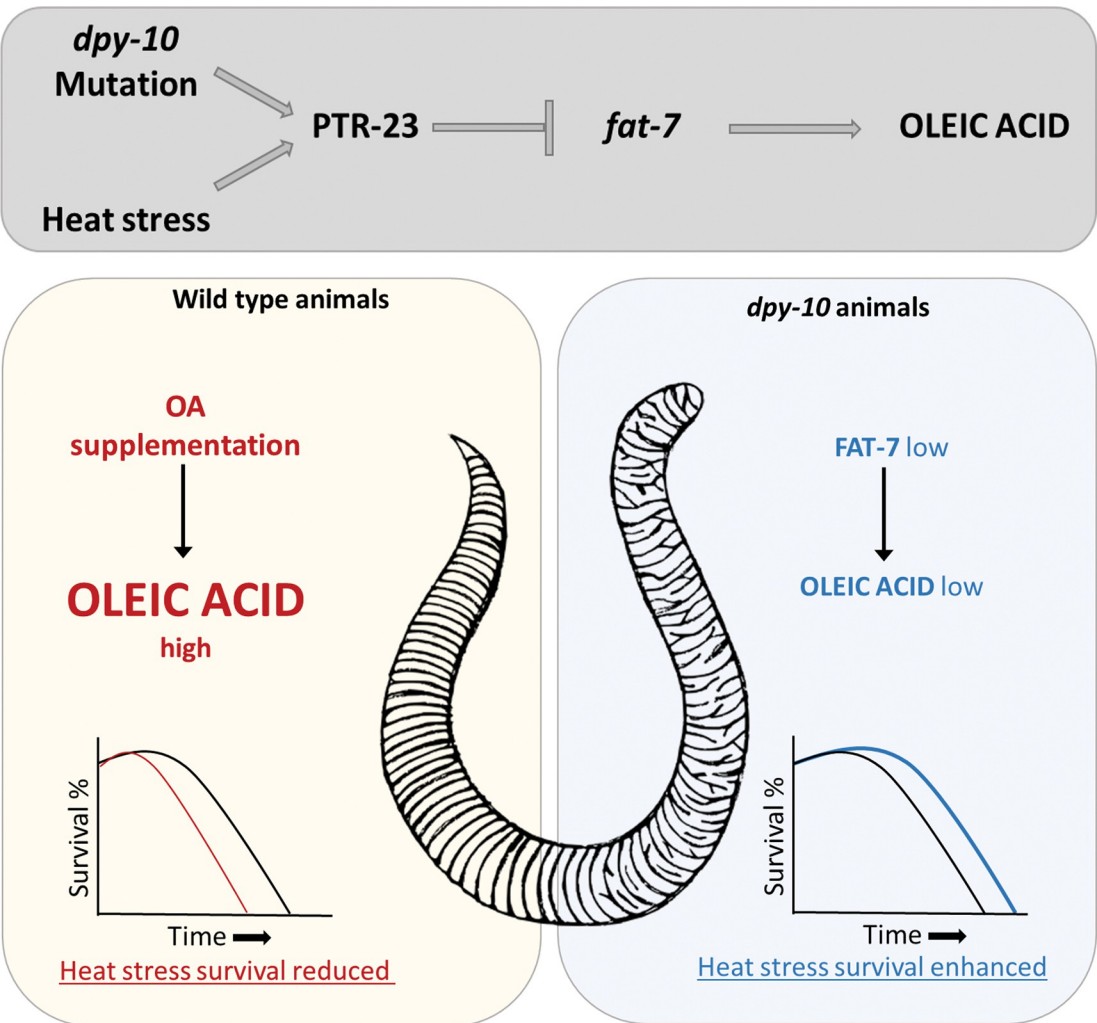

**Fig 7. Model for heat stress adaptation in *C. elegans*.** Heat stress or cuticle defect leads to transcription of patched-like receptor *ptr-23*. PTR-23 upregulation leads to the downregulation in the transcription of the fatty acid desaturase, *fat-7*. Lower levels of FAT-7 result in lower oleic acid levels, thus promoting the survival of worms during heat stress.

WT and show a dramatic increase in OA [47]. Therefore, the reduced survival of *fat-2* is attributable to the accumulation of OA in these mutants, consistent with the survival of FAT-7 OE and OA-supplemented animals. Notably, upon OA supplementation or FAT-7 OE, *fat-2* is upregulated (Fig 2E–2F), suggesting a feedback regulatory role for FAT-2 in regulating the levels of OA. During heat stress, we see a downregulation in the levels of *fat-7* transcripts, while the *fat-2* transcripts are upregulated. Overall, our results suggest that excess OA is problematic for survival during heat stress, and the WT animals actively try to balance OA levels by downregulating *fat-7* and upregulating *fat-2*.

Why do the levels of unsaturated fats need to be altered during heat stress? Heat stress is known to increase membrane fluidity, possibly leading to altered localization and functioning of membrane-associated proteins. In this scenario, the fine-tuning of membrane fluidity using various combinations of saturated and unsaturated fats should improve the functioning of membrane proteins. Saturated fatty acids are densely packed and hence decrease fluidity in the membrane. Consistent with the expectation that saturated fats should increase and

unsaturated fats should decrease during heat stress, we observe a decline in the expression of desaturases (*fat-1, -3, -4, -5, -6,* and *-7)* in heat-stressed worms. This is concomitant with an overall increase in various saturated fatty acids and a decrease in unsaturated fatty acids in the GC-MS data. A previous study showed that the knockdown of *fat-6, fat-7,* and elongase gene *elo-2* by RNAi increased saturated fatty acid content and thermal stress resistance. Whereas, it had an inverse effect on the survival of these animals against exogenous toxins like paraquat [12]. We found that overexpression of *fat-7* or oleic acid supplementation reduces *C. elegans* survival on heat stress but does not affect survival of *C. elegans* on NaCl hyperosmotic stress and hydrogen peroxide-induced oxidative stress. This suggests that lipidomic changes are crucial to modulate membrane fluidity for adaptation to heat stress but might be less important for other stresses.

Changes in the extracellular matrix due to chemical or shear stress can cause disruption of the plasma membrane [52], possibly triggering cytosolic stress response. In our previous study, we observed that PD collagen mutants show enhanced heat stress resistance [48]. Disruption of the cuticle in worms could lead to mechanical stress, activating stress response in the plasma membrane. Indeed, we find that disruption of the cuticle in PD collagen mutant, *dpy-10*, alters desaturase expression. *dpy-10* animals show modest upregulation of desaturases *fat-1* and *fat-4* and a large decrease in *fat-7* transcript levels without heat stress. This suggests that alteration in the extracellular matrix due to PD collagens defect could trigger epidermal membrane remodeling. Levels of saturated fatty acids (C14:0 and C16:0) are higher, while levels of unsaturated fatty acids (C18:2, C20:4n-3, and C20:5) are reduced in *dpy-10* animals compared to WT. Concomitant with the downregulation of *fat-7* transcript, oleic acid (C18:1) levels were reduced in *dpy-10* animals compared to WT animals. *dpy-10* animals showed an increase in levels of two PUFAs (C20:3 and C20:4), unlike heat-stressed WT worms. This suggests that the nature of membrane remodeling due to cuticle defects is somewhat different from heat stress-induced membrane remodeling. Unlike cuticle disruption, heat stress triggers the expression of small heat shock proteins, which likely assist in proteostasis as well as membrane remodeling as chaperones, contributing to these differences.

Here, we find that patched-like receptor PTR-23 is essential for survival under HS in WT animals as well as cuticle-defective *dpy-10* mutants. PTR-23 appears to be the nodal point where the response to heat stress and cuticle defects impinge. PTR-23 activity is central for the downregulation of FAT-7 desaturase expression in WT as well as *dpy-10* animals. *ptr-23* animals show altered levels of saturated fatty acids as well as some unsaturated fatty acids compared to WT animals (S1 Table). Consistent with an increase in *fat-7* expression in *ptr-23* mutants, oleic acid levels are also higher compared to WT animals. The *ptr-23; dpy-10* double mutants have higher oleic levels and increased *fat-7* expression compared to *dpy-10* worms. This suggests that defects in the extracellular matrix and heat stress modulate cuticular signaling to regulate lipid metabolism.

PTR-23 (patched-related protein), a transmembrane protein, is a homolog of the *Drosophila* Patched (Ptc) protein. Ptc proteins are receptors for the Hh ligands and regulate developmental signaling events via the Hh signaling in *Drosophila* and vertebrates [53, 54]. Interestingly, *C. elegans* lacks all Hh signaling components except *Ptc*, suggesting that the nematode lacks canonical Hh pathway [40]. However, the worm encodes several patched receptors, some of which are known to perform non-canonical functions. Catillo *et al.* reported that PTC-3, a C. *elegans* patched homolog, regulates lipid homeostasis and controls cellular cholesterol levels by acting as a cholesterol transporter [41]. Patched receptors share homology with bacterial RND transporters, and PTR-4, in particular, may act as a transporter of hydrophobic molecules, suggesting that patched/PTR proteins are transmembrane pumps for hydrophobic cargos [55]. PTR-23 might also be a lipid transporter. However, we show that PTR-23

functions in the epidermis to non-cell-autonomously regulate FAT-7 expression, which is not consistent with the idea of a transporter. Furthermore, the lack of hedgehog signaling components in *C. elegans* suggests the possibility of non-canonical interacting partners for PTR-23, that facilitate this skin-gut axis of fat regulation. Notably, PTR-23 has been shown to modulate osmo-sensitivity in *C. elegans* [50]. This patched receptor, expressed in the cuticle, negatively regulates the mucin-like protein OSM-8 to regulate osmolarity-related genes and osmo-sensitivity. Here, we show an additional non-canonical role for PTR-23, where the transmembrane receptor negatively regulates *fat-7* to orchestrate adaptation to heat stress. PTR-23 functions in the epidermis to regulate lipid metabolism via *fat-7*, which is expressed primarily in the gut and the skin. Interestingly, FAT-7 is known to be regulated by transcription factors, such as NHR-49, SBP-1, HLH-30, NHR-64, and NHR-80, and co-activator MDT-15. We believe that PTR-23 might impinge onto these known regulatory networks of FAT-7 to affect its expression during heat stress [23, 24, 26, 27, 29, 30, 44, 56, 57]. Our results highlight a non-canonical role for PTR-23 or at least its repurposing in *C. elegans*. The presence of a skin-gut axis to regulate stress response in adult worms suggests that similar mechanisms might also operate in other multicellular animals.

## Methods

### Strains and reagents

*C. elegans* strains used in the study were wild-type N2 (Bristol), *CB88 [dpy-7(e88)]*, *CB1281 [dpy-8(e1281)]*, CB12 *[dpy-9(e12)]*, CB128 *[dpy-10(e128)]*, VSL2003 *[dpy-10(ves2003)]*, VSL2004 *[dpy-10(ves2003); ptr-23(ok3663)]*, VSL2007 *[dpy-10(ves2003); ptr-23(ok3663); fat-7:: gfp (nIs590)*, VSL2014 *[dpy-7(e88); fat-7::gfp (nIs590)]*, VSL2015 *[dpy-8(e1281); fat-7::gfp (nIs590)]*, VSL2016 *[dpy-9(e12); fat-7::gfp (nIs590)]*, VSL2017 *[dpy-10(e128); fat-7::gfp (nIs590)]*, VC3219 *[ptr-23(ok3663)]*, DMS303 *(nIs590 [fat-7p::fat-7::GFP + lin15(+)] V)*, BX150 *(waEX18 [fat-5::GFP + lin15(+)])*, NR222 (kzIs9 [(pKK1260) lin-26p::NLS::GFP + (pKK1253) lin-26p::rde-1 + rol-6(su1006)]) and VP303 (kbIs7 [nhx-2p::rde-1 + rol-6 (su1006)]). All strains were maintained at 20°C. Reagents used in the study are paraquat (methyl viologen dichloride hydrate, Sigma, cat#856177), hydrogen peroxide (Sigma, cat# H1009), sodium oleate (Sigma, cat#O7501), and sodium linoleate (Sigma, cat#L8134). *dpy-10* deletion strain VSL2004 was created by CRISPR/Cas9 genome editing as described previously (Dickinson & Goldstein, 2016). 25 ng/μl of sgRNA (GCTCTACCATAGGCACCACG) against *dpy-10* was microinjected with 25 ng/μl Cas9 plasmid (a gift from Prof. K. Subramaniam, IIT-Madras) in N2. Dpy phenotype animals were picked and further checked for deletion in *dpy-10* by PCR (FP: gtcgtcgttacctcagcca, RP: CAAGTCACCTTCTGGAGTTG) followed by sequencing. *dpy-10(ves2003); ptr-23(ok3663)* double mutant was created by crossing *dpy-10 (ves2003)* hermaphrodites with *ptr-23(ok3663)* (FP-ttctgccaaatcaatgaaccag, RP-TCTAG-CAAAGGAGAATCCAG) males as per standard protocol. DMS303 has an extra copy of *fat-7*:: GFP and hence has been used for both reporter and OE study [31]. NR222 and VP303 were used to induce tissue-specific RNAi in the epidermis and intestine, respectively.

### RNA interference

For systemic RNA interference by feeding, we grew empty vector control and target gene dsRNA expressing HT115 (DE3) in LB broth containing 50 g/ml carbenicillin overnight at 37°C [58, 59]. These cultures were plated on NGM plates containing 5 mM isopropyl-D-thio-galactoside (IPTG) and 50 μg/ml carbenicillin. After 24 hours at 25°C, these plates were used for synchronization of worms of various genotypes.

## Survival assays

To induce chronic heat stress, 25–30 synchronized adult worms per genotype were transferred to freshly seeded *E. coli* OP50 plates and maintained in a 32˚C incubator [60]. Animals were scored for survival every 4–6 hours. For single time point heat stress assay, animals were scored only once after 24 hours of incubation at 32˚C. For oleate supplementation assays, NGM plates supplemented with 0.8 mM sodium oleate (Sigma-Aldrich) were prepared as described earlier [61], and worms were grown on oleate supplemented plates. For hydrogen peroxide assay, 35 mm plates were prepared with 2 ml of 1.5% agarose 30 mins prior to the experiment and dried at RT. 10 mM hydrogen peroxide solution was prepared in M9 buffer, and a volume of 2 ml was added to each plate. Age synchronized 25–30 adult worms were transferred to each plate and scored for survival every 1 hour for 6 hours at RT. For single time point hydrogen peroxide experiment, animals were scored after 6 hours of incubation in 10 mM hydrogen peroxide at RT.

## Paralysis assay

For hyperosmotic stress, 10–15 adult animals were placed on NGM plates containing 500 mM NaCl and no food [62]. Each genotype was tested in duplicates. For the survival curve, paralysis was scored every minute for 10 minutes. For single time point experiments, paralysis was scored at 10 minutes.

## Statistical analysis

Survival graphs were analyzed using the software GraphPad PRISM 5.01 (Kaplan-Meier method). Survival curves with $p$ values $<0.05$ based on Mental-Cox test were considered significantly different. Bar graphs were analyzed using Student's $t$-test, where $p$ values $<0.05$ were considered significant with one-on-one comparisons. In cases with multisample comparisons, a post-hoc Tukey or a post-hoc Dunnet test was performed, where $p < 0.05$ was considered significant. Survival statistics are presented in S2 Table.

## Quantitative Real-time PCR

Synchronized adult animals were exposed to 20˚C or 32˚C for 4,8 or 12 hours. Animals were harvested using M9 buffer in a 15 ml falcon tube. Excess of M9 buffer was removed, and worms were frozen at -80˚C in 1 ml of TRizol. RNA was extracted using the RNeasy Plus Universal Mini Kit according to the manufacturer's instructions (Qiagen). cDNA was prepared using the iScript cDNA synthesis kit (BIO-RAD). qRT-PCR was conducted using the BIO-RAD TaqMan One-Step Real-time PCR protocol using SYBR Green fluorescence (BIO-RAD) on an Applied Biosystems QuantStudio 3 real-time PCR machine in 96-well plate format. Fifty nanograms of RNA were used for real-time PCR. 10 μl reactions were set up in two technical replicates and performed as outlined by the manufacturer (Applied Biosystems). Relative fold changes were calculated using the comparative *CT* $(2^{-\Delta\Delta CT})$ method and normalized to *act-1* [63]. Three or more biological replicates were used for qRT-PCR analysis.

## Imaging

Synchronized population of adult FAT-5::GFP and FAT-7::GFP [64] animals were exposed either to 20˚C or 32˚C and imaged after 4 or 8 hours. Animals were placed in 1 mM Sodium Azide on 2% agarose pads and imaged at 5X or 10X magnification. Imaging was done using a SIRION ultrahigh-resolution microscope. For each experiment, at least 10 worms were imaged. Three or more biological replicates were done for the analysis.

## Gas chromatography-mass spectrometry (GC-MS/MS)

GC-MS/MS to analyze fatty acid composition was done as previously described [21]. Briefly, ~400 synchronized adult animals with 0–8 embryos were collected and washed twice with water on ice. Excess water was removed, and worms were incubated with 2 ml of 2.5% sulfuric acid in methanol for 1 hour at 70˚C to convert fatty acids to respective methyl esters. After incubation, 1 ml of water was added to stop the reaction, and 200 μl of hexane was added to solubilize and extract fatty acids esters. 2 μl of the hexane layer was added onto an Agilent 7,890 GC/5975C MS in scanning ion mode equipped with a 20 × 0.25 mm SP-2380 column (Supelco) to quantify the relative levels of fatty acid esters.

## Hoechst staining

Hoechst staining to detect cuticle permeability was done as previously described [48]. 30–40 adult worms were maintained at 20˚C and transferred to 32˚C (HS) or continued to be maintained at 20˚C (Ctrl) for 24 h. The worms were washed thrice and resuspended in M9 containing Hoechst stain as previously described [48]. After 30 minutes of staining, the worms were washed thrice with M9 and Imaged with a DAPI filter using an epifluorescence microscope.

Source data is available on dryad [65].

## Dryad DOI

https://doi.org/10.5061/dryad.15dv41p3j [65]

## Supporting information

**S1 Fig. FAT-5 levels are downregulated upon heat stress.** (A) FAT-5::GFP expression in ctrl and HS animals at indicated times and temperatures (scale bar, 250 μm) and their (B) quantification. Statistical significance for (B) was calculated using a post-hoc Dunnett test. Ns (not significant), * $P \leq 0.05$; ** $P \leq 0.01$; *** $P \leq 0.001$; **** $P \leq 0.0001$.
(TIF)

**S2 Fig. Heat shock response entails alteration in enzymes of fatty acid metabolism.** qPCR analysis of (A) *fat-1*, (B) *fat-2*, (C) *fat-3*, (D) *fat-4*, (E) *fat-5*, (F) *fat-6*, (G) *fat-7*, (H) *dgat-2*, (I) *acs-2*, and (J) *hsp-16.2* in HS (32˚C) over ctrl (20˚C) WT animals. ns, non-significant; *$P < 0.05$; **, $P < 0.01$; ***, $P < 0.001$; ****, $P < 0.0001$ as determined by unpaired '*t*' test with Welch's correction. Error bars represent SEM.
(TIF)

**S3 Fig. FAT-7 OE and Oleic acid supplementation does not affect survival during other abiotic stresses.** (A) Survival of WT and FAT-7 OE animals during osmotic stress of 500 mM NaCl. (B) Survival of WT and FAT-7 OE animals during acute oxidative stress upon 6-hour exposure to 10 mM $H_2O_2$. (C) Kaplan-Meier survival curves of WT and WT supplemented with oleic acid during osmotic stress upon exposure to 500 mM NaCl ($P = 0.7329$; N = 25–30; n = 3). (D) Kaplan-Meier survival curves of WT and WT supplemented with oleic acid during oxidative stress upon exposure to 10 mM $H_2O_2$ ($P = 0.6384$; N = 25–30; n = 3). ns (not significant).as determined by unpaired Student's '*t*' test. Error bars represent SEM.
(TIF)

**S4 Fig. Heat shock causes permeabilization of *C. elegans* cuticle.** DIC and DAPI filter images of WT adults grown for 24 h at (A) 20˚C or (B) 32˚C followed by staining with Hoechst stain. (C) Percentage of worms permeabilized as quantified by Hoechst staining in the region of interrest (ROI) around the pharynx. ***, $P < 0.001$ as determined by unpaired '*t*' test with

Welch's correction. Error bars represent SEM.
(TIF)

**S5 Fig. PTR-23 regulates heat stress resistance of *dpy-10* mutant.** (A) Kaplan Meier survival curves of WT and *ptr-23* animal with *dpy-10* RNAi at 32˚C (P<0.0001; N = 25-3-; n = 3). Kaplan-Meier survival curves of N2 and *ptr-23(ok3663)* animals during (C) oxidative stress (P = 0.5671; N = 25–30; n = 3) and (D) osmotic stress (P = 0.6442; N = 25–30; n = 3). (D) Kaplan Meier survival curves of VP303, intestinal RNAI animals with *ptr-23* RNAi (P = 0.5906; N = 25–30; n = 3). (E) Transcript levels of fatty acid desaturase genes in *ptr-23 (ok3663)* mutants compared to WT animals. *P < 0.05; **, P < 0.01; ***, P < 0.001 as determined by unpaired '*t*' test with Welch's correction. Error bars represent SEM.
(TIF)

**S1 Table. Fatty acid analysis by Gas chromatography-mass spectrometry.**
(DOCX)

**S2 Table. Statistics for Survival assays.**
(DOCX)

## Acknowledgments

Some *C. elegans* strains were provided by the CGC, which is funded by the NIH Office of Infrastructure Programs (P40 OD01440).

## Author Contributions

**Conceptualization:** Siddharth R. Venkatesh, Ritika Siddiqui, Anjali Sandhu, Jennifer L. Watts, Varsha Singh.

**Formal analysis:** Siddharth R. Venkatesh, Ritika Siddiqui, Anjali Sandhu.

**Funding acquisition:** Varsha Singh.

**Investigation:** Siddharth R. Venkatesh, Ritika Siddiqui, Anjali Sandhu, Malvika Ramani, Isabel R. Houston, Jennifer L. Watts.

**Methodology:** Siddharth R. Venkatesh, Ritika Siddiqui, Anjali Sandhu, Jennifer L. Watts, Varsha Singh.

**Supervision:** Varsha Singh.

**Visualization:** Varsha Singh.

**Writing – original draft:** Siddharth R. Venkatesh, Ritika Siddiqui, Anjali Sandhu, Jennifer L. Watts, Varsha Singh.

**Writing – review & editing:** Siddharth R. Venkatesh.

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
