## [Decision Letter · Decision Letter 0]

2 Nov 2023

Dear Dr Singh,

Thank you very much for submitting your Research Article entitled 'Homeostatic control of stearoyl desaturase expression via patched-like receptor PTR-23 ensures the survival of C. elegans during heat stress' to PLOS Genetics.

The manuscript was fully evaluated at the editorial level and by independent peer reviewers. The reviewers appreciated the attention to an important topic but identified some concerns that we ask you address in a revised manuscript.

We therefore ask you to modify the manuscript according to the review recommendations. Your revisions should address the specific points made by each reviewer.

Yours sincerely,

Danielle A. Garsin

Academic Editor

PLOS Genetics

Gregory P. Copenhaver

Editor-in-Chief

PLOS Genetics

Reviewer's Responses to Questions

**Comments to the Authors:**

Reviewer #1: Venkatesh et al present their findings on the role of PTR-23 and certain fatty acids, specifically oleate, in the resistance to heat stress. They propose a model where PTR-23 inhibits the expression of the fatty acid desaturase fat-7, which decreases the amount of monounsaturated oleic acid, promoting animal survival during exposure to high temperature. These findings are of interest as they demonstrate a non-canonical role for the hedgehog signaling pathway, or at least it’s repurposing by C. elegans. The manuscript is well-written and easy to follow.

I only have a few small suggestions / comments:

ptr-23(RNAi) worms are more sensitive to heat shock. Is this specific? I.e. what about hyperosmotic stress that was used as a non-specific control for fat-7 mutants or fatty acid supplementation?

How does PTR-23 inhibit expression of fat-7? I understand that addressing this experimentally would be beyond the scope of the manuscript, but can the authors add to the discussion section their ideas for the signal transduction pathway, particularly given that C. elegans does not have full hedgehog signaling pathway present.

Labels in Figure 1C are too small. For the pathway diagram to be useful, we should be able to see and read what is there.

Are the GC-MS data for WT at 20C in Fig 1C and Fig 3A actually exactly same data? Were all the samples for the MS collected at the same time, in parallel?

Do not use strain names (e.g. NR222) in legends. It makes it harder to understand what is going on. Instead use gene names.

While overall the manuscript is well-written, there are some typos that need to be taken care of. For example, the first sentence of the abstract: “Organismal responses to temperature fluctuations includes an evolutionary conserved cytosolic chaperone machinery as well as…” Should be corrected to “responses … include” (i.e., correct agreement in plurality).

Reviewer #2: Venkatesh et al. PLoS Genetics 2023

In this work, Venkatesh et al. demonstrate that the spectrum of saturated and unsaturated fatty acids changes in heat stress animals. They demonstrate that the MUFA oleic acid and the enzyme FAT-7 that synthesizes it reduce heat resistance. They also analyze links between fatty acid desaturases and heat stress in permeability-defective dpy-10 mutants and mutants in Patched receptor-related PTR-23. The experiments convincingly show coordination between fatty acid desaturation and heat stress resistance. The conclusions about the relationship of PTR-23 to these components are less well supported.

Major comments

1. Double mutant analyses presented suggest that PTR-23 and DPY-10 act independently to regulate heat stress response, because the heat resistance of dpy-10;ptr-23 is intermediate between those of dpy-10 and ptr-23, rather than demonstrating epistasis. The best evidence that ptr-23 acts downstream of dpy-10 is the regulation of ptr-23 transcripts in the dpy-10 mutant. The two conclusions could be reconciled by proposing other target genes downstream of dpy-10 that also contribute to its phenotype.

2. Figure 5 is missing the panel with lipid analysis after heat shock.

3. The authors should consider the possibility that PTR-23 acts as a lipid transporter rather than as a receptor, an interpretation that has been presented in recent papers. An issue with the receptor model is that a receptor should act cell autonomously in the tissue in which it is expressed, but this study demonstrates that it does not.

Minor comments/typos

Title: I’m not sure that “Homeostatic” is the best description for this type of control, because it implies maintenance of a condition, rather than a stress response.

Abstract: “includes an evolutionary” should be “include an evolutionarily”

What is meant by “physiologically relevant” heat stress? Are these conditions that would be expected in the natural habitat?

Fig. 1B How long is the heat stress condition that was applied in this experiment (fat gene RT-PCR)?

Fig. 2E,F are not discussed in the text

p. 10-11 The authors show that heat stress causes permeability defects in the cuticle. This result is presented as supporting the conclusion that permeability defective dpy-10 mutants cause defects in heat stress, but I find that it puts the direction of causality into question. Because of this circular causation, a simpler explanation is that permeability and heat stress response are jointly regulated by something else. Can the authors discuss their ideas for how/why heat stress causes permeability defects in the context of their model?

p. 11 “no effect of PTR-23 on other desaturases” According to the supplemental data, there are significant differences in expression of the other desaturases.

p. 11 “cutilcle” should be “cuticle”

p. 14 “suppression of FAT-7 expression” would be more accurately worded “repression of FAT-7 expression”

p. 14 “To study the impact of PTR-23 on oleic acid levels, we studied the levels of fat-7” is not consistent. Do the authors mean “To study how PTR-23 impacts oleic acid levels”?

Fig. 6 title “PTR-23 regulates fatty acid levels” does not describe the experiments shown.

Fig. 6 – It would be helpful to have statistics that compare heat shock to control for each genotype (rather than comparisons of genotypes under the same conditions).

Reviewer #3: The study by Venkatesh, Siddiqui, Sandhu, and colleagues investigated how lipid desaturation contributes to the response of C. elegans to heat stress. Specifically, the study focused on whether unsaturated fatty acids, like oleic acid, negatively affect survival during prolonged stress. The authors find that decreasing the levels of polyunsaturated fatty acids by suppressing desaturase enzymes plays a crucial role in the worms' response to heat stress. Accordingly, excessive oleic acid is harmful to survival under chronic heat stress conditions.

The study also explored the connection between cuticle collagen defects and lipid balance within the worm's body. In heat-stress-resistant dpy-10 worms, the expression of fat-7 is decreased, leading to a decrease in unsaturated fatty acids and an increase in saturated fatty acids. The authors further discovered that the heat stress resistance of the dpy-10 mutant is regulated by the epidermal patched-like receptor PTR-23, which serves as a negative regulator of fat-7 expression.

The topic of this paper is of great and broad interest. In particular this study provides new insight into the role of an epidermal patched-like receptor and thus of a C. elegans epidermis to intestine axis in heat stress adaptation. The presented results are mostly (see my comments below) convincing and often validated by different experimental approaches (e.g. using single and double mutants and tissue-specific RNAi for functional analyses, supplementation experiments for determining the role of oleic acid, or gas chromatography-mass spectrometry for assessing fatty acid composition in various genetic backgrounds). My main reservation concerns the presentation of the survival data, since it does not become clear if the graphs that are shown are based on data from multiple or single experimental runs and if any statistical analysis was done to determine P-values . I found the survival statistics in Table S2 eventually, but I strongly recommend to present the data in a way that the reader can understand everything from the figure legend (also see my comments below).

Minor comments:

line 88: delete ‘by’

line 128/129: I suggest to refer to Fig 1C in line 125-126 (and omit this sentence)

line 136: Please mention the gene name (dgat-2)

line 141: What is meant by ‘exquisitely’ in this context?

line 213: Please replace ‘are linked’ by ‘is linked’

line 216/217: Was this an unexpected finding? Since DPY-10 is a permeability-determining collagen and heat-stress causes permeability defects, it is not obvious why mutations in dpy-10 lead to enhanced survival during heat stress.

Survival data in e.g. Fig 2 A-D, Figure 4 C, E, and F, Figure S5 A-D: Please add sample size (i.e. 25-30 worms according to methods section) and number of experimental runs to figure legend. Do the Kaplan-Meier survival curves show a representative run or pooled data? How many experimental runs were done? Where do I find the raw data? The graphs show the Kaplan-Meier survival curves, but was any statistical analysis done to determine P-values?

Fig 4: In the legend, please explain the NR222 strain, so that the reader does not have to look into the main text to understand what was done.

Fig 7: The worm looks rather like an earthworm.

**Have all data underlying the figures and results presented in the manuscript been provided?**

Reviewer #1: Yes

Reviewer #2: **No: **Figure 5 is missing a panel.

Reviewer #3: **No: **I could not find any raw data.

PLOS authors have the option to publish the peer review history of their article (what does this mean?). If published, this will include your full peer review and any attached files.

Reviewer #1: No

Reviewer #2: No

Reviewer #3: No

---

## [Editor Report · Decision Letter 1]

15 Nov 2023

Dear Dr Singh,

We are pleased to inform you that your manuscript entitled "Homeostatic control of stearoyl desaturase expression via patched-like receptor ensures the survival of C. elegans during heat stress" has been editorially accepted for publication in PLOS Genetics. Congratulations!

Yours sincerely,

Danielle A. Garsin

Academic Editor

PLOS Genetics

Gregory P. Copenhaver

Editor-in-Chief

PLOS Genetics

Comments from the reviewers (if applicable):

**Data Deposition**

http://datadryad.org/submit?journalID=pgenetics&manu=PGENETICS-D-23-01054R1

**Press Queries**

---

## [Editor Report · Acceptance letter]

4 Dec 2023

PGENETICS-D-23-01054R1 

Homeostatic control of stearoyl desaturase expression via patched-like receptor PTR-23 ensures the survival of *C. elegans* during heat stress 

Dear Dr Singh, 

We are pleased to inform you that your manuscript entitled "Homeostatic control of stearoyl desaturase expression via patched-like receptor PTR-23 ensures the survival of *C. elegans* during heat stress" has been formally accepted for publication in PLOS Genetics! Your manuscript is now with our production department and you will be notified of the publication date in due course.

With kind regards,

Zsofi Zombor

PLOS Genetics

On behalf of:
